# Designing a Waterless Toilet Prototype for Reusable Energy Using a User-Centered Approach and Interviews

**Hyun-Kyung Lee**

Division of General Studies, UNIST, Ulsan 44919, Korea; hlee@unist.ac.kr

**Abstract:** User-oriented community engagement can reveal insights into ways of improving a community and solving complex public issues, such as natural resource scarcity. This study describes the early process of co-designing a novel, waterless toilet to respond to the water scarcity problem in the Republic of Korea. It presents how we designed a toilet focusing on three factors—a sanitization function, an ergonomic posture, and clean aesthetics—by conducting focus group interviews as part of a user engagement approach to understand what community users want from a toilet and ways of improving their toilet experiences. The results not only supported the development of an experiential service design project to raise community awareness of water scarcity but also supported scientists and engineers in experimenting with and developing new technologies by collaborating closely with designers.

**Keywords:** design prototype; toilet design; collaboration with scientists; interdisciplinary convergence; natural resource; feces

## 1. Introduction

Community engagement is a suitable approach for solving complex public issues, such as natural resource scarcity. It may also reveal what people already know about an issue and provide insights that could support the creation of a better natural resource community. This study describes the fuzzy front end (FFE) stage of designing a novel, waterless toilet prototype, the first collaborative research project between designers and scientists to respond to the water scarcity problem in South Korea. The aim of this collaboration was not only to raise community awareness of water scarcity, but also to save water. The co-design of the waterless toilet at the FFE followed a community-engaged research approach that influenced the technology push and supported "silent design": that is, design decisions made by non-designers [1]. This collaboration project sought to understand what users want in a toilet and to improve the toilet experience. Unpredictably, the approach led to the creation of an experiential service design project to raise community awareness of water scarcity.

## 2. Background: Water Scarcity

Water is one of the most vital resources for human viability. However, there is not enough for the needs of everyone in the globe. Due to population growth, many countries face water shortages; thus, they need to prepare for future problems [2]. Water scarcity also results in low economic growth. In 2012, the Organisation for Economic Co-operation and Development (OECD) recommended that both OECD and non-OECD countries reduce water risks, such as water shortages (including droughts), inadequate water quality, non-accessible water, and anything that undermines the resilience of freshwater. Yet, all countries face different water scarcity problems.

According to the United Nations Environment Programme [3], the rapid economic growth and high population density of the Republic of Korea have caused water shortages and freshwater scarcity. This has long been—and still remains—a critical challenge. Reports indicate that current aqua-ecosystem protection mechanisms are insufficient [3]. The Ministry of Land, Transport and Maritime Affairs (2011) reported that only 26% of water in Korea is fit for use and that each Korean person uses only one-sixth of the global average for individual water consumption. Hence, the nation is considered a water-stressed nation.

Water in the Republic of Korea is contaminated for three reasons: (1) damage to the nation's hydro-ecology caused by construction near rivers, which has decreased limnobios, animals, and freshwater plants; (2) soil runoff; and (3) the eutrophication of rivers due to an increasing inflow of phosphorus into rivers [4]. According to the previous literature, people consider treated wastewater from sewage treatment plants to be clean; however, attached algae are widely visible, largely due to phosphorus discharged from farmlands during the summer flood season. More critically, the shallow depth and slow current speed of the river increase detention time, increasing nutrient salts during annual drought seasons. Since phosphorus is not filtered from sewage, it grows too much. It originally comes from toilets, and some toilets are installed in areas that lack sewage. Thus, it is important to remove phosphorus from toilets. This can be accomplished by a sewage treatment plant.

In 2015, the Ministry of Science, ICT (Information and Communications Technologies), Future Planning (MSIP), initiated the Convergence Research Centre (CRC) project, a cross-disciplinary research project that connects the arts, design, and science to prepare for future sustainable energy. Solving the water scarcity problem is one of several seven-year research projects. The collaboration team comprised two design researchers, one philosopher, seven different disciplinary scientists (e.g., water, climate change, and urban environment), and industry partners (e.g., different groups of natural artists, digital artists, graphic designers, and engineers). The team is planning to build a temporary research lab to research these issues at the UNIST (Ulsan National Institute of Science and Technology) in the Republic of Korea. The research lab building will be two stories high and approximately 99 m$^2$.

## 3. Waterless Toilet

A family of four uses an average of 225 L of water per day. This represents 27% of household water consumption [5]. Domestic houses use 65.9% of total water [6] and represent one of the main causes of contaminated water in Korea. To remove phosphorus, water scientists from the Department of Urban and Environmental Science initiated a waterless toilet project.

The conventional sanitation system, the flush toilet, has improved public health and protected humans from waterborne diseases. The "drop-and-discharge" approach is a convenient and comfortable solution for disposing human excreta because it rapidly removes hazardous and unpleasant matter from households [7]. However, the complete disconnect between people and their excreta causes environmental problems. People discharge their excreta unconsciously, meaning that our environment has been forced to receive the massive flow of contaminants. In particular, algal blooming and red-tide problems occur in many countries, even developed countries with well-established wastewater treatment systems. Waterless toilets that dispose of excreta in a sanitary way could reduce the high level of clean water usage. Returning sanitized dried feces to the land may also improve the soil environment. Thus, a waterless toilet team—one of the CRC project teams—has begun to investigate alternative toilet systems. The team has explored ways to develop a prototype to examine the effectiveness of eliminating phosphorus from the fuzzy front end (FFE) stage, which is considered the earliest stage of the new product development process [8].

A prototype can be a tool for solution generation or evaluation, as well as a vehicle for team collaboration [9,10]. Supporting explorations that allow users to innovate new patterns of use in the field and over lengthy periods may provide designers insights into the domestication of radically new concepts [11].

## 4. User-Centered Research Approach

### 4.1. The FFE Process

This project is characterized by a radical technology push, such that engineers in the field develop a set of research routines based on their beliefs about what is feasible or worth trying [12–14]. However, according to NESTA [15], so-called "front-end" research into technological development, market trends, and consumer needs are now staples in design industry operations. Numerous studies identify communication as critically important because this phase allows for modifications, reorientations, and radical changes in new product planning [16]. FFE is the least expensive stage of a project [16]. This collaboration team, therefore, communicated once every two or three weeks regarding how a prototype, by developing technology to remove phosphorous from a waterless toilet, should be designed. In order to clarify the product's concept during the FFE stage, which is known to be complicated (most companies fail to have clear product definitions [17]), we reviewed the literature on new product development (NPD) during the FFE stage. Based on a theory of marketing, operation management, and engineering design, Krishnan and Ulrich define product development as "the transformation of a market opportunity and a set of assumptions about product technology into a product available for sale" [18]. Similarly, Ulrich and Eppinger define product development as "the set of activities beginning with the perception of a market opportunity and ending in the production, sales, and delivery of a product" [19]. A successful product may be viewed as a solution to customers' demands [20]. In order to design elegant and efficient solutions to customers' needs, two different types of information must be combined: "need information" (what users need) and "solution information" (how products are built) [20]. The National Endowment for Science and Technology and the Arts (NESTA) has proposed that researching people's needs, tastes, and preferences is critical for shaping new products and services [15]. However, though consumer needs should be the focal point of any design process, many organizations neglect them [21]. As a result, many design processes fail to reach their target consumers or end users [22,23]. In addition, due to the rapidity of modern technological development, many users find it difficult to directly specify their needs regarding a particular product, service, or experience [24,25]. For this reason, Borja de Mozota recommends employing user-centered and informed design research methods to support the development of products and services from the beginning so not to fail to find out what users need [26].

To design something that is meaningful for users, it is important to understand users' experiences, perceptions, and ways of making sense of things [27]. Having a thorough understanding of one's target users and their needs is necessary to be successful. Mossberg found that the growth of industry through customer experience is reflected in new patterns of consumption, new demands, and new technology [28]. He insisted that a customer's experience is a blend of many elements that have come together to involve consumers emotionally, physically, intellectually, and spiritually in an ethnographic approach. Similarly, Pine and Gilmore propose that combining NPD with customer experience creates more profit than applying NPD on its own because, in the combined approach, every touch point with the customer serves as a market opportunity [29]. Several successful organizations that regard customer experience as an aspect of economic value have been evaluated [30]. These companies position experience as a planned journey with multiple touch points, such that the packaged experience defines the characteristics of a product, service, or brand [31]. Taking the above into consideration, we aimed to identify and learn about community members who would be potential future users of our waterless toilet. Specifically, we followed a reflective problem-solving approach [32] to design prototyping. Figure 1 presents how user-centered research influenced the design of our waterless toilet.

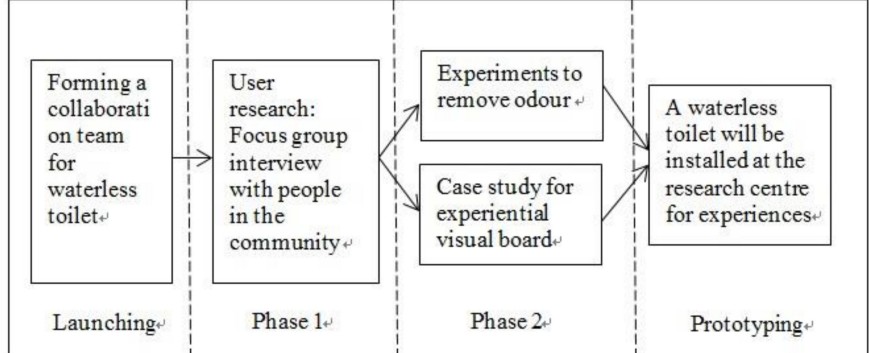

**Figure 1.** Co-designing a waterless toilet using the fuzzy front end (FFE) process.

### 4.1.1. Phase 1: Focus Group Interviews

We conducted focus group interviews to investigate user perceptions and understandings of the water scarcity problem and a waterless prototype.

#### Sampling Criteria

The study aimed to understand the people in the UNIST campus community where the test-bed research lab will be built. A total of 54 participants (15 females and 39 males)—all of whom had used public toilets and toilets in their homes and who would be potential users of the waterless toilet prototype—were involved in the focus group interviews. There were nine groups, each of which comprised six UNIST undergraduate students majoring in different areas of engineering (e.g., life science, chemical engineering, physics, computer engineering, electronic engineering, mechanical engineering, and engineering management).

#### Process

To ensure that all of the key issues were covered, semi-structured interview questions were used. The participants were asked about (1) their awareness of water scarcity in Korea, (2) their previous experiences attempting to save water, (3) how they could be motivated to save water, (4) their perceptions of a waterless toilet, and (5) their dream toilet. They were also asked to share their thoughts and ideas about how to save water. All interviews were audio-recorded and recorded via memo form that I used in collecting and writing about my observations, which were later analyzed using thematic analysis.

#### Results

Though the participants in each group had different majors, all shared common ideas and perceptions about each question. The thematic analysis revealed three themes: a disbelief about water scarcity, a motivation to save water, and an experience requirement for the waterless toilet.

(1) Disbelief about water scarcity: Everyone had heard that the Republic of Korea lacked water since they were children. However, most did not think that this was true, since the water quality in Korea is good and since they could access sufficient water to meet their needs anytime and anywhere. One individual thought that the push to save water (i.e., the many water-saving media campaigns) was propaganda. Only a few people had actually tried to save water through such methods as "using a cup to brush their teeth," "filling a water basin to wash their faces," or, in the case of one individual, "putting a plastic bottle in the toilet water tank to flush less water."

(2) Motivation for saving water: Most participants said that they had not attempted to try to save water because they did not believe that the Republic of Korea was truly suffering from water scarcity. The participants had no direct experiences of water scarcity, and most mentioned that, since their water bills were not expensive, they were not aware of how much water they

used each day. They were from a demographic that was less likely to experience shortage or poor quality of water. As possible motivators to save water, the participants recommended water usage indicators or visual indicators of water consumption. They also suggested that more media exposure would be helpful in raising awareness of water scarcity. Lastly, they commented that financial losses would motivate them to save water. Some participants wanted tax deductions for saving water.

(3) Experience requirement for the waterless toilet: This theme emerged from the two questions about the participants' perceptions about a waterless toilet and their dream toilets. None of the participants had previously thought about their dream toilet. However, they said that they wanted a new toilet based on their public and private toilet experiences: visual cleanness, sanitariness (including automatic cleaning around the inside of the bowl), automatic flushing after usage, no bad odor, soundproofing within a public toilet, and a comfortably warm seat cover. Since the participants had not previously considered the water scarcity problem, they explained ways in which they would improve the current style of toilet. They also said that they would try a waterless toilet if they came across it in a public restroom. Most asked for simple visual guidance on how to use a waterless toilet, which is designed to suck up feces like a "vacuum cleaner" and send it directly to the energy production system. It requires about half liters of water, which is significantly less than what a regular toilet consumes.

The results of the focus group interviews revealed that, although people had heard of the water scarcity problem, they had never before tried to save water because they could always access water easily and at their convenience. A design activity exposes new issues and information needs as the work progresses [33,34]. The aim of designing the waterless toilet prototype was to improve the situation water scarcity; however, if the community is not aware that there is a genuine water scarcity problem, they are less likely to accept a waterless toilet. Therefore, since the participants required a realistic experience to motivate them to save water, we chose to design not only a waterless toilet but also an experiential visual guideline that indicates the water scarcity problem. Moreover, the engineering team decided to remove all negative odors from the toilet. This decision was based on the user-centered approach and was considered a sociable design, or a design "for the benefit of the people who use it, taking into account their true needs and wants" (Norman 2010, 130). Thus, the co-design of the waterless toilet, which was led by a technology push, was divided into two projects to meet the needs of the people in the community who would become the future users of the toilet and participants in solving the water scarcity problem.

4.1.2. Phase 2: Case Studies for Service Design Guideline

The aim of case studies is to study good practices of visual and experiential designs capable of leading and increasing civic engagement and awareness of the need to save energy.

Criteria

In order to identify case studies capable of meeting the experiential aims of civic engagement by providing realistic visual experiences, we contacted the Korea Institute of Design Promotion (KIDP), a non-profit organization for design support under the Ministry of Trade, Industry and Energy. Though there is no service design related to water scarcity, the design policy experts and staff of the KIDP recommended three service design cases for the public sector in Korea, all involving the design and management of public campaigns to reduce energy bills and raise awareness of the need to save energy. These studies explored (1) which visual elements were strategically designed to lead or motivate civic engagement to reduce energy usage and (2) which approaches and principles were applied to the design experience to increase people's engagement.

(1) Redesigning an apartment energy bill (http://www.slideshare.net/sdnight/ss-30524771). This case concerned one of the first service designs for a utility service in Korea. It was conducted

on an apartment town comprising 600 apartments in Bangbae Dong, Seoul, and its results were highly effective, reducing the total energy bills by 10%. Many redesigning projects graphically change a certain part of an energy bill; however, in this case, the designers and design researchers reduced the energy bill based on in-depth interviews with the community. This case study prompted other apartment towns to accept similar guidelines.

(2) National Health Insurance Service (http://www.slideshare.net/usableweb/ss-16567992?related=1). This case involved redesigning a health check-up chart for the National Health Insurance Service at Ilsan, Myungji Hospital. The chart had been criticized for being difficult to read by people who were not medical doctors from specific areas. In other words, doctors in one medical center often struggled to understand examinations conducted in other medical centers. The redesigned check-up chart received a 95% satisfaction rating.

(3) Changwon National Industrial complex. As foreign labor for industrial complexes has increased, many industrial safety accidents have occurred. To reduce safety accidents, which can also waste energy resources (e.g., in cases of hydrofluoric acid leaks), a user-centered approach was employed. The service design involved changing signage and installing pictograms for international laborers. This case was published online (http://economy.hankooki.com/lpage/industry/201504/e20150406173204120170.htm).

Process

Secondary data of visual campaign were collected through online case design websites (www.designdb.com) and suggested website addresses. Thematic analysis was used to identify each case's key characteristics in relation to creating civic engagement experiences, reducing energy usage, and motivating people to do things they had not done before.

Case Study Results

Though all three cases were different, they shared similar characteristics in that all employed a user-centered approach to encourage people to reduce their energy usage or do things in which they had not previously been involved. Two themes emerged.

(1) Easy readability. Most people were interested in their health conditions or how much they were charged on their utility bill. However, previous bills had been designed with small font sizes and logos or advertisements. Based on user interviews, both bills were redesigned to use larger fonts and exclude or deemphasize less important information.

(2) Vivid color for nudging. Both the apartment bill case and the industrial complex case used a vivid red color for warnings. If an individual used more energy than that used by the average household nationwide, he or she received a red-colored bill that was visible to every neighbor. If an individual used less energy than the national average, he or she received a green-colored bill. Average utility usage generated a yellow-colored bill. Being able to compare their consumption with that of their neighbors motivated many users to reduce their consumption. Similarly, in the case of the national industrial complex, toxic pipelines were colored red to alert international laborers who did not know Koreans to be cautious. This step reduced workplace accidents. In short, using a vivid color and improving readability increased competitiveness among residents and safety among laborers.

*4.2. Influence of User Research on the Waterless Toilet Design*

These case studies made it clear that a user research approach reveals opportunities to improve civic engagement. Mayhew and Bias indicated that such an approach supports higher returns in terms of increased product value [35]. The case study participants actively participated to reduce utility bills and safety accidents. They seemed to be tuned into the effort to overcome problems caused by a lack of staff [36]. Moreover, Boyle and Harris indicated that collaborating with users could guarantee that

services meet their requirements. Thus, user research supports better outcomes and encourages active user engagement in self-help and positive behavioral change, thus avoiding possible challenges in the future.

In order to engage with people in the community (a research test bed at UNIST), we used focus group interviews to determine community members' needs for realistic experiences related to water scarcity and for vivid visual representations to notify/warn about utility usage, health insurance check-ups, and safety cautions. To raise awareness about water scarcity in the community, the collaboration teams recommended encouraging people to visit the research lab to examine the waterless toilet prototype. Ramaswamy and Gouillart indicated that the participatory approach could improve stakeholder engagement, and could lead to higher productivity, higher creativity, and lower costs and risks [37]. Thus, the researchers created a visual board as a service design, which served as a channel for experience prototyping [38]. After the prototype at the research lab was developed and designed, this visual board was used to support community members in discussing their experiences and perceptions on water scarcity. As the community members visited the lab, it was possible to monitor their different behaviors in a real context using several variations of observation-based ethnographic field methods [39]. It was also possible to involve the users in designing and delivering services to achieve the full benefits of co-creation [39].

## 5. Prototyping: Waterless Toilet Design

In designing the toilet, we focused on three factors: the sanitization function, an ergonomic posture, and cozy aesthetics.

Sanitization Function

First, from the focus group interview, we drew the conclusion that people want a clean toilet experience. Thus, we designed the prototype for sanitization (sterilization) and cleanliness. We installed ultraviolet (UV) lights on the cover so that the toilet looks sanitizable. These UV lights also disinfect and sterilize the surface of the seat and the bowl of the toilet (Figure 2). Additionally, since the word "toilette" means "dressing room" in French, the seat cover was designed to resemble a dressing room vanity chair, which was intended to make people feel warm. The toilet hip area was designed with enough space for anyone to sit comfortably. Ultimately, the toilet was designed to provide not only a clean experience, but also a restful one.

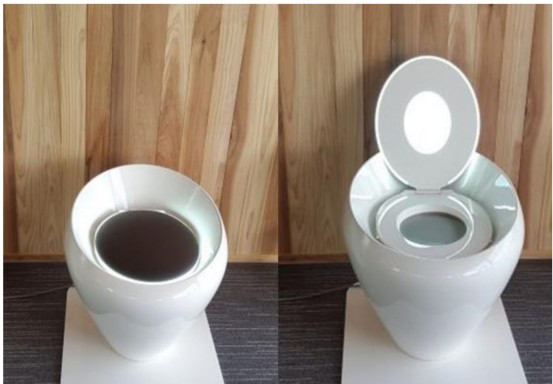

**Figure 2.** Prototype of the waterless toilet.

Ergonomic Posture

Second, the toilet was designed in an ergonomic manner to support proper posture for defecating. The seat can be adjusted (raised or lowered) to an appropriate angle to fully relax the muscle around the colon and support quick and easy defecation. Once a user sits down on the toilet, the back side of seat tilts slightly, raising the user's knees. This encourages users to maintain an optimal posture

for relaxing the muscle around the colon. The concept for this posture came from the Paimio Chair, designed by Alvar Aalto (Figure 3) for patients in a tuberculosis sanatorium in Finland. When the patient sits down and leans on the chair, the patient's posture changes the angle of the chair, helping the patient breathe better and comfortably arranging his or her body organs.

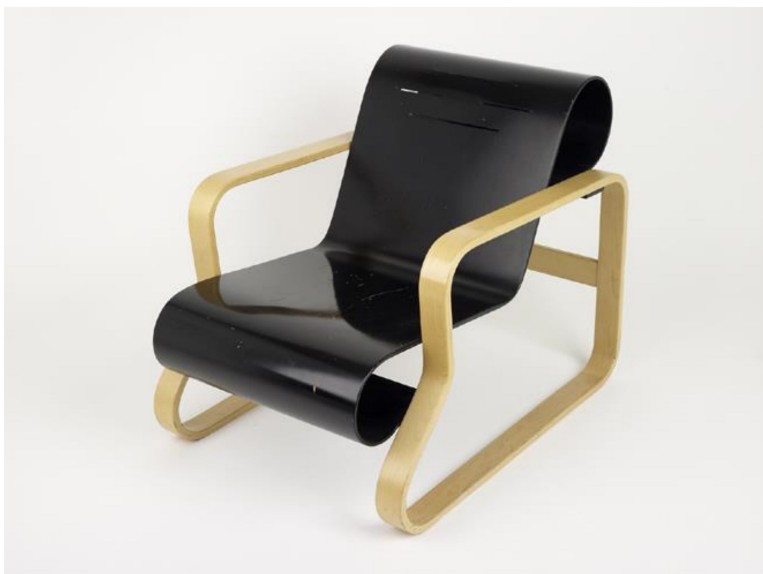

**Figure 3.** The Paimio Chair designed by Alvar Aalto in Finland.

The waterless toilet is also designed ergonomically with respect to the angle of the sitting posture. Although the tilted seat relaxes the muscle around the colon, it also makes it difficult for users to stand up (compared to the upright posture of a traditional toilet). Though this may not pose a significant problem for young and healthy people, it could encumber older, disabled, or injured people. Therefore, we installed a spring under the back side of the seat that users can activate by pushing back slightly when they need to stand up. Figures 4 and 5 present suboptimal and optimal posture angles defecation.

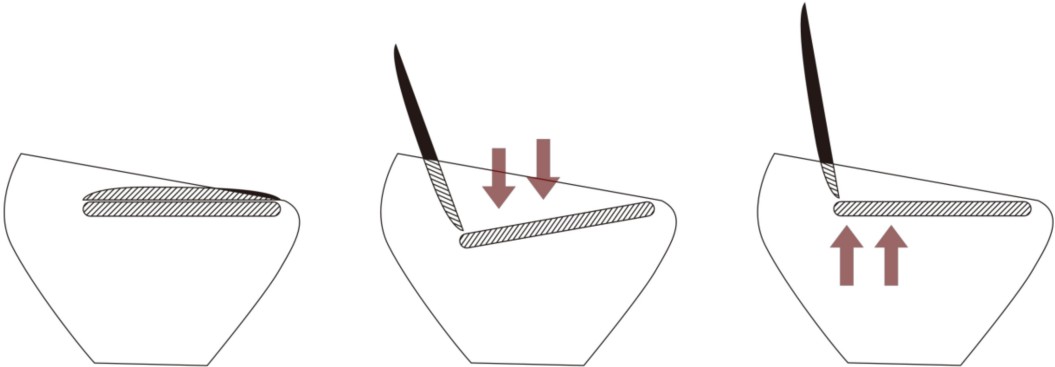

**Figure 4.** Working process of the seat of the waterless toilet prototype.

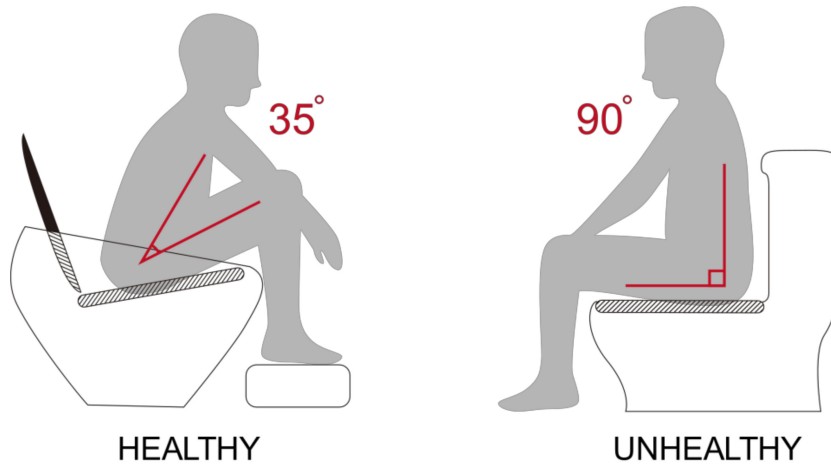

**Figure 5.** The proper posture and angle to relax a human's intestines.

Cozy Aesthetics

Thirdly, the appearance of the toilet prototype was inspired by the white porcelain of the Yi Dynasty (Korean imperial household, called the Joseon household) (Figure 6).

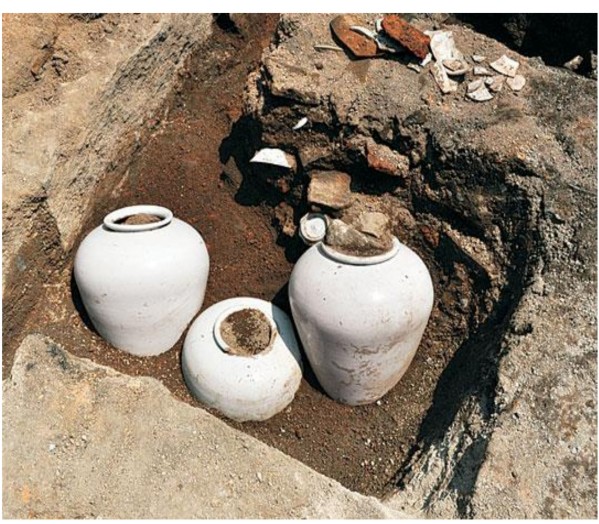

**Figure 6.** Korean Yi Dynasty porcelain.

Toilet shapes have not changed very much since their first development in the 16th century. However, this new type of toilet offers several improvements. First, unlike a traditional toilet, which has a tapered bowl shape to support a suction mechanism located at its bottom back side, our proposed toilet does not require centrifugal force or a vacuum system to suction the feces. Therefore, the waterless toilet can be slimmer or wider. We chose a shape that was cozy and familiar. Second, the white porcelain of the Yi Dynasty suggests familiar characteristics of interior objects that already play a role in our everyday lives. Figure 6 presents the aesthetics of the waterless toilet. Users think this is a great idea that has Korean tradition and culture embedded in it.

## 6. Conclusion

This article describes the FFE process of designing a waterless toilet to combat the problem of water scarcity in the Republic of Korea. The waterless toilet prototype was co-designed based on user perceptions, a technology push, and the use of a radical technology to remove phosphorus from toilets. Through focus group interviews, we identified several key factors regarding water scarcity

and users' perceptions of a waterless toilet: (1) disbelief concerning the state of water scarcity in Korea, (2) a motivation to save water, and (3) experience requirements concerning the waterless toilet.

We observed that people in Korea generally do not seriously consider Korea's water scarcity problem. However, if they experience water scarcity visually and realistically, they may be more likely to try to save water. We also found that people want a toilet to be sanitary, odorless, clean, and comfortable. These findings helped to reveal what users want and need to improve their toilet experience, and they guided the direction of the prototype development to be more user-oriented and less rushed (i.e., due to the technology push).

Based on our user research, we designed our waterless toilet focusing on three main factors: a sanitization function, an ergonomic posture, and cozy aesthetics. First, based on the focus group interview, we concluded that people want a clean toilet experience. The prototype was therefore designed with UV lights on its cover for sanitization (sterilization) and cleanliness. Second, the toilet was designed in an ergonomic manner to allow people to sit in a posture appropriate for defecating. Specifically, the seat can be lowered and adjusted to the optimal angle to relax the muscle around the colon. Thirdly, the appearance of the toilet prototype was inspired by the white porcelain of the Yi Dynasty and includes familiar characteristics of interior elements and everyday objects.

The researchers' user-centered and community-focused approach helped this collaborative research project gain new information for further improvements. The approach used in this study can be employed as a guideline for researching and solving other public problems. Engaging people in the community supports the identification of both current and future public problems. Furthermore, responding to potential users' opinions, recommendations, and insights encourages positive participation and thus a better public environment. Further studies evaluating the waterless toilet prototype with the aid of community members are needed to improve the toilet experience and to increase awareness of the current situation of water scarcity in Korea.

**Author Contributions:** H.-K.L. conceived and designed the experiments and wrote the paper.

**Acknowledgments:** This work was supported by the National Research Foundation of Korea (NRF) Grant funded by the Korean Government (MSIP) (No. NRF-2015R1A5A7037825).

**Conflicts of Interest:** The author declares no conflict of interest.

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
