# Peer review of "Designing a Waterless Toilet Prototype for Reusable Energy Using a User-Centered Approach and Interviews"

_applsci, doi:10.3390/app9050919_

Round 1

Reviewer 1 Report

I have attached a file with many sticky notes in the pdf which will hopefully help the authors.

This paper has much of interest in it and I think it should be published after some revisions. Here are my concerns:

The introduction and literature reviewing is repetitive in places while in others references cited are just lists of names without showing real use of the literature to create the context for the study. The authors could rethink this resulting in a shorter but more useful introduction. It would be helpful for the general reader to give some indication of what sort of "waterless toilet" is at issue here as well.

There needs to be some comment on the demographic of the focus group sample to indicate if the fact that they were all university students is likely to affect the result.

The themes identified in the focus groups should be supported with selected quotations from the recordings (anonymized, obviously).

Some illustrations would assist the discussion on visual engagement and an earlier explanation of what this even is would assist the general reader. It becomes clear in the end.

After the analysis of the various examples, we enter an extended narrative section which does not flow as well as the earlier part of the paper. How we get from the focus groups via the analysis of visual engagement to the design of the toilet could be thought through and structured better.

I was concerned with the sideways leap into physical experiments on drying faecal matter. On reflection I think this would be better omitted altogether as a description and just referenced as a research report or similar as only the conclusion that it is possible to reduce odour really matters to the main argument of this paper. The description of the experiments is a distraction.

Sadly what we do not get is the "what the users thought of the finished design" which I was expecting all the way through but which did not come - presumably it is the intention to publish this later. I can live with this but the conclusions which are drawn need to be tightened up. My understanding is that they are i) university attending groups don't believe there is a water scarcity; ii) visual campaigns can be effective and the authors have designed one around water scarcity using a toilet design as a feature; iii) an analysis of user consultation indicated that certain features were important in a toilet and the finished design attempts to address these in specific ways such as posture control and aesthetic design.

If this could be made really clear and perhaps some consideration given to the stated aims and objectives at the beginning so that the conclusions tie to them then the paper will be publishable.

Finally a note on ethics: my concerns are minor but some statement should be made about the anonymous nature of the focus groups and the way data on individuals was/is held, and on the ethical and safety issues of experimenting on human facaes (if this part of the paper is not removed as I suggest).

Author Response

Thank you so much for your considerate and kind requests for improving my paper.

I amended everything you suggested and mentioned. You will see them in the attached word file.

For example, "Cozy -> Clean, spends -> uses, excreted -> discharged, overbreeds -> grows too much, memo form that I used in collecting and writing about my observations" your suggestion is all accepted and changed. 

Thank you so much for your help!!

Reviewer 2 Report

Dear author,

- All the while the sanitation sector could do with some serious design development and user-involvement, which is the strength of this paper, those elements need to be deeply anchored within the field they are trying to influence. I don't see this in this paper, see my broad comments below, and detailed ones in the attached document.

- There is a crucial need for you to team up with someone that in depth understands the wastewater sector, eutrophication, water scarcity in order to correctly frame the research. The text as it stands now is confusing, contradictory, not using the right terminology and making sweeping statements that are in the best of cases correct but unsubstantiated and in the worst of cases simply incorrect.

- A great idea to draw on national design experiences for changing behavior outside the field of water scarcity, but for full understanding of how to frame it within the field of water scarcity I would have expected you to also draw on international design experiences actually tackling behavioral change within the specific field of water scarcity - there must be such experiences and it needs to be related to.

- Figure 2 is deeply puzzling and deeply disturbing. I cannot understand what the point of the drying experiment with coffee grounds really is trying to represent. This and that you relate the figure to odor, are the puzzling elements. The disturbing element is that you present a figure of data drawn on two samples in a scientific journal. Experiments need to be set up in a scientific manner, to allow for replicability and to aim for statistically significant/non-significant results. Anything less should not make its appearance in a scientific journal. I suggest that section gets removed completely or, if it is of any, although to me incomprehensible, importance, the experiments need to be redone and better explaine with appropriate methods section accompanying.

- You have NOT designed a waterless toilet! You seem to have made a shape and mechanism for sitting down, and some kind of UV sanitization mechanism but you have said nothing about how the excreta is conveyed without water through your toilet! Furthermore you have NOT removed phosphorus from the toilet, you have not even explained how the excreta is supposed to be transported through your system, how it is to be handled or anything. You seriously need to understand that to be able to claim to have designed a waterless toilet that removes P you need to explain the technical functions behind these two very technical claims. I don't see any of this in this paper.

-You MUST relate your results to the existing body of knowledge - you have to make an effort to read up on the sanitation field, existing design efforts (there's been a few not the least due to Reinvent the toilet by BMGF) within the field etc. When one ventures outside one's field one has to do so with respect for existing knowledge, and try to further that, at least when one wants to publish in a scientific journal. 

- This manuscript is not fit for publication without a really major MAJOR overhaul, preferably by you as an author to team up with some sanitation sector colleagues. You need to identify objectives of the study, and orientate the text and results accordingly. As it is now the text is all over the place, going back and fourth between different objectives (saving water, "removing P", saving energy) - there is no red thread through the manuscript. Moreover, it is really disturbing that you present results in a figure based on 2 samples, and the results are somehow not at all related to the rest of the text. 

Author Response

(The authors gave the same response as above.)
